# Identification of phenothiazine derivatives as UHM-binding inhibitors of early spliceosome assembly

Pravin Kumar Ankush Jagtap [1,2,7], Tomáš Kubelka [1,2,7], Komal Soni[1,2], Cindy L. Will[3], Divita Garg[1,2], Claudia Sippel[4], Tobias G. Kapp[2,5], Harish Kumar Potukuchi[1], Kenji Schorpp[1], Kamyar Hadian [1], Horst Kessler[2,5], Reinhard Lührmann[3], Felix Hausch [4,6], Thorsten Bach [2]✉ & Michael Sattler [1,2]✉

Interactions between U2AF homology motifs (UHMs) and U2AF ligand motifs (ULMs) play a crucial role in early spliceosome assembly in eukaryotic gene regulation. UHM-ULM interactions mediate heterodimerization of the constitutive splicing factors U2AF65 and U2AF35 and between other splicing factors that regulate spliceosome assembly at the 3′ splice site, where UHM domains of alternative splicing factors, such as SPF45 and PUF60, contribute to alternative splicing regulation. Here, we performed high-throughput screening using fluorescence polarization assays with hit validation by NMR and identified phenothiazines as general inhibitors of UHM-ULM interactions. NMR studies show that these compounds occupy the tryptophan binding pocket of UHM domains. Co-crystal structures of the inhibitors with the PUF60 UHM domain and medicinal chemistry provide structure-activity-relationships and reveal functional groups important for binding. These inhibitors inhibit early spliceosome assembly on pre-mRNA substrates in vitro. Our data show that spliceosome assembly can be inhibited by targeting UHM-ULM interactions by small molecules, thus extending the toolkit of splicing modulators for structural and biochemical studies of the spliceosome and splicing regulation.

[1] Helmholtz Zentrum München, Ingolstädter Landstr. 1, 85764 Neuherberg, Germany. [2] Center for Integrated Protein Science Munich (CIPSM), Department Chemie, Technische Universität München, Lichtenbergstrasse 4, 85748 Garching, Germany. [3] Max Planck Institute for Biophysical Chemistry, Cellular Biochemistry, Am Fassberg 11, 37077 Göttingen, Germany. [4] Max Planck Institute of Psychiatry, Kraepelinstr. 2–10, 80804 Munich, Germany. [5] Institute for Advanced Study (IAS), Technische Universität München, Lichtenbergstrasse 4, 85747 Garching, Germany. [6] Institute for Organic Chemistry and Biochemistry, Technische Universität Darmstadt, Alarich-Weiss-Strasse 4, D-64287 Darmstadt, Germany. [7] These authors contributed equally: Pravin Kumar Ankush Jagtap, Tomáš Kubelka. ✉email: Thorsten.Bach@ch.tum.de; sattler@helmholtz-muenchen.de

P re-mRNA splicing is a key regulatory step in gene expression, which removes non-coding introns from pre-mRNAs to form mature mRNAs suitable for translation. The process is performed with single nucleotide precision and errors in the process lead to aberrant splicing being the cause of several human diseases including cancer progression[1–4]. Moreover, alternative splicing (AS) is an essential mechanism that greatly expands the coding capacity of eukaryotic genomes by generating multiple protein isoforms from a single primary transcript and provides an additional layer of gene regulation[5,6]. The regulation of AS involves the recognition of *cis* regulatory elements, i.e., short RNA sequence motifs, by *trans* acting factors, i.e., RNA binding proteins. In addition, many protein-protein interactions also play important roles in the regulation of constitutive and alternative splicing [7–14].

During complex E formation in the early stages of pre-mRNA splicing, the U2AF65-SF1 complex recognizes the consensus sequences on the pre-mRNA near the 3′ splice site (3′ss)[15–19]. The protein-protein interaction between U2AF65 and SF1 is mediated by the C-terminal U2AF homology motif (UHM) domain of U2AF65 and the N-terminal UHM ligand motif (ULM) of the SF1 protein[8]. UHM-ULM interactions mediated by multiple ULMs of SF3b1 and UHM domains from SPF45, PUF60 and CAPER-α play a crucial role in AS (Supplementary Fig. 1)[8,10,20]. UHM domains were first identified in both subunits of the U2AF heterodimer (U2AF65 and U2AF35). Structural analysis revealed that they represent non-canonical RNA recognition motif (RRM) domains with degenerate RNA binding motifs[19,21]. UHM domains comprise an Arg-X-Phe amino acid sequence (where X can be any amino acid), located in the loop connecting the α-helix B and the β-strand of the UHM domain. Also, the α-helix A has a more acidic character than the canonical RRM domain[22].

In recent years, UHM-ULM interactions have been identified in several other proteins including KIS kinase and MAN1 which mediate diverse biological functions[8,10,20,22–24]. However, the role of UHM-ULM interactions in these proteins is not completely understood. Several UHM containing proteins have been associated with human diseases. For example, SPF45 activates a cryptic 3′ss in β-thalassemia and is overexpressed in breast, lung, colon, and ovarian tumors[25]. Mutations in splicing factors U2AF1 and SF3b1, which mediate UHM-ULM interactions have been associated with myelodysplastic syndromes and myelofibrosis[26–29].

Compared to transcription and translation, there are only a few well-characterized inhibitors available to study splicing mechanisms and spliceosome assembly, such as spliceostatin and its related compounds, isoginkgetin, 1,4 naphthoquinones, 1,4-heterocyclic quinones and cp028[30–36]. An impressive example of using small molecules[37–39] or antisense oligonucleotides[40] for therapy are the recently developed splicing modulators of the SMN2 gene linked to the severe spinal muscular atrophy (SMA) disease, which led to the FDA approval of the first therapy against SMA using antisense oligonucleotides.

Only few inhibitors are known to act during the early phase of spliceosome assembly, i.e., the formation of the early E complex, an important stage for the regulation and fidelity of constitutive and alternative splicing[41,42]. As UHM-ULM interactions are recurring in early spliceosome assembly and have been structurally well characterized, these interactions provide an ideal target for the development of inhibitors that modulate early spliceosome assembly. The emerging role of these proteins in diseases underscores the importance of understanding the role of UHM-ULM interactions in these processes. We recently developed cyclic peptides derived from the natural linear ULM peptide ligands and could show that they are able to inhibit early spliceosome assembly[43]. As trafficking of peptides across the cell membrane is challenging, we wished to identify small molecule inhibitors of UHM domains. Using high-throughput screening (HTS) of a highly curated library of 43296 drug-like molecules with biophysical assays monitoring binding to UHM domains, we identified phenothiazine derivatives as inhibitors of UHM-ULM interactions. These small molecule inhibitors target a highly conserved tryptophan binding pocket on the UHM domains. Using structure activity relationship (SAR) studies we characterized the functional groups in these inhibitors required for UHM binding and structurally characterized the interaction of these inhibitors using NMR and X-ray crystallography. In vitro splicing assays indicate that the inhibitors target early spliceosome assembly, presumably by inhibiting the UHM-ULM interactions.

## Results

**Identification of small molecules targeting UHM-ULM interactions.** In order to develop a fluorescence polarization (FP) assay for UHM-ULM interaction, we tagged a cyclic peptide, which in our earlier study showed high-binding affinity and similar binding mode to the SPF45 UHM domain as that of native ULM[43] with fluorescein dye ([sc,sc(KSRWDE)]-K-C-fluorescein; (Fig. 1a)). This peptide maintained its binding affinity of 1.46 μM as measured by isothermal calorimetry (ITC) (Fig. 1b). It also showed a concentration dependent binding to the SPF45 UHM domain in the FP assay and showed an $EC_{50}$ value of 2.8 μM (Fig. 1c).

The FP assay was used to screen a library of 43296 compounds at a fixed concentration of 125 μM in duplicate (Fig. 1d). Hits were defined as compounds that show a decrease in the polarization value by three standard deviations from the mean plate value and were reproducible in duplicates. From the initial screening we identified 57 compounds, which were further validated by multipoint titrations in an FP assay and in $^{1}$H-$^{15}$N HSQC NMR titrations. Out of the 57 compounds, 7,8-dihydroxyperphenazine showed binding in FP assays and in protein-observed $^{1}$H-$^{15}$N HSQC NMR titration experiments and was thus chosen for further studies.

**Structure activity relationship (SAR) studies with 7,8-dihydroxyperphenazine.** As 7,8-dihydroxyperphenazine is prone to oxidative degradation due to the presence of hydroxyl groups in the 7- and 8-positions of the phenothiazine ring it was not suitable for further structural studies[44]. We performed SAR studies with 7,8-dihydroxyperphenazine derivatives in order to assess the importance of functional groups for binding to the UHM domain and also to obtain a compound with suitable affinity for structural studies. For determining the affinity of the compounds for SPF45, we used an Amplified Luminescent Proximity Homogeneous Assay (AlphaScreen) using His-tagged protein as donor and biotin tagged native ULM peptide as acceptor in the assay[45]. In this assay, 7,8-dihydroxyperphenazine showed an $EC_{50}$ value of 10.4 μM (Fig. 2a). Breaking of the central ring of the phenothiazine by removing the sulfur atom (Cmp1) or removal of substituents on the phenothiazine ring (Cmp2 and Cmp3) led to a drastic decrease in binding affinity. Substitution of the piperazine moiety by an imidazole or benzimidazole group (Cmp4 and Cmp5, respectively) reduced the affinity fourfold. However, the addition of a single methoxy group at the 7-position in perphenazine (Cmp6) led to a ≈ 6.5-fold gain in affinity compared to Cmp2 and Cmp3. Replacing the two hydroxyl groups with methoxy groups maintained the affinity at around 10 μM (Cmp7) (Fig. 2b, c).

From these SAR studies we concluded that the phenothiazine ring mediates key interactions with the UHM domain. In addition, substitutions of the phenothiazine ring were required

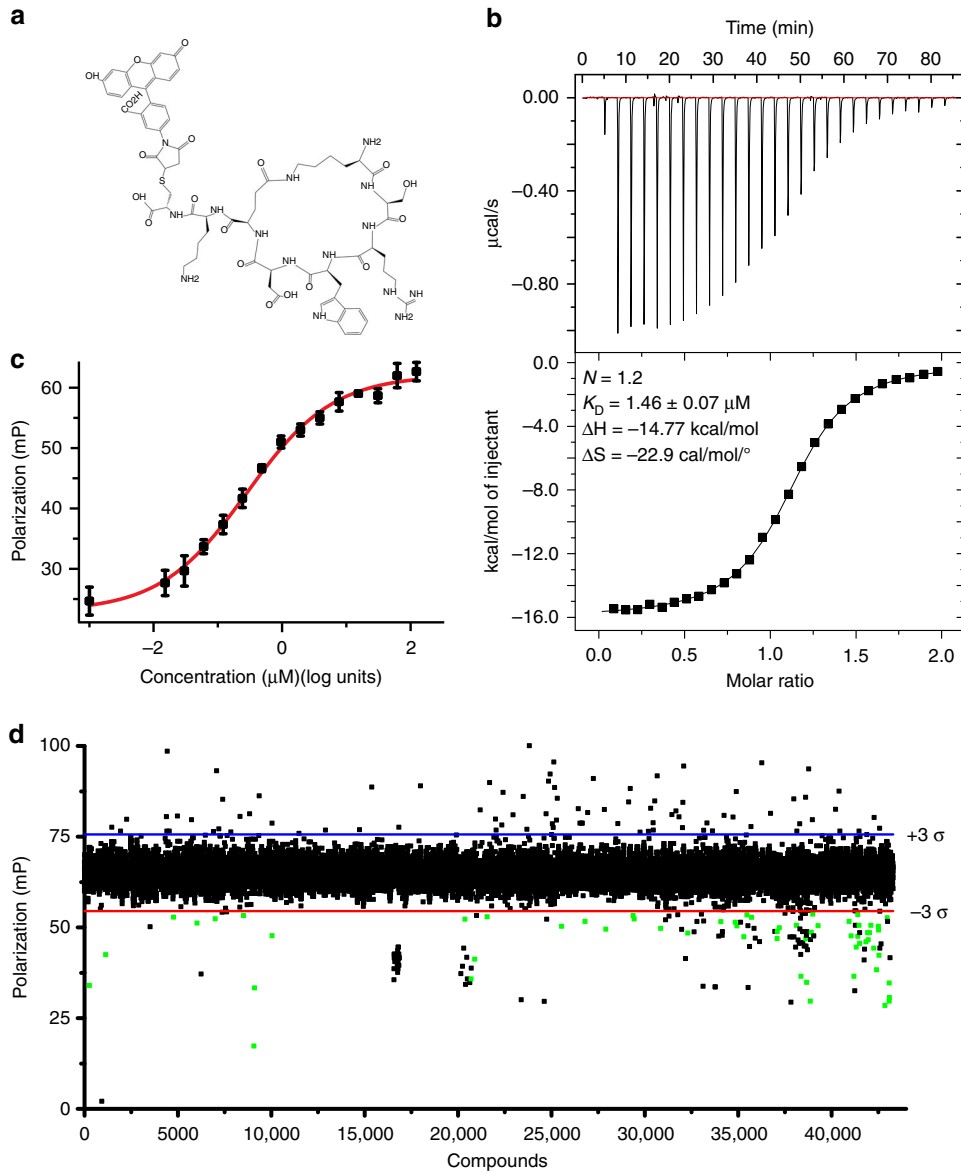

**Fig. 1 HTS to identify inhibitors of UHM-ULM interaction. a** Cyclic peptide tagged with fluorescein, [sc,sc(KSRWDE)]-K-C-fluorescein, used as tracer in HTS. **b** ITC shows that the binding affinity of the interaction between the SPF45 UHM domain and [sc,sc(KSRWDE)]-K-C-fluorescein is $1.46 \pm 0.07\,\mu M$. Error represents fitting error. **c** A fluorescence polarization assay shows an $EC_{50}$ value for the interaction of $2.8 \pm 0.4\,\mu M$. Error bars show standard deviation of three technical triplicates and error is calculated as fitting error ($n = 3$). **d** Results of the 1st HTS campaign are shown. Compounds which showed a threefold standard deviation decrease from mean value in the intensity of the polarized light and were reproducible in a duplicate assay are highlighted as green dots ($n = 2$).

for maintaining the affinity of the compound indicating that they also take part in the interaction with the UHM domain. As addition of a single methoxy group at the 7-position in Cmp6 significantly restored the affinity, this methoxy presumably interacts with the SPF45 UHM domain via polar or electrostatic interactions. Also, as the hydroxyl groups in 7,8-dihydroxyperphenazine can be replaced by methoxy groups, these hydroxyl groups apparently do not act as hydrogen bond donors involved in hydrogen bonding interactions with the residues in the UHM domain.

**Binding site mapping of the phenothiazine inhibitor on the SPF45 UHM domain.** To determine the binding site of 7,8-dihydroxyperphenazine and 7,8-dimethoxyperphenazine (Cmp7) on the SPF45 UHM domain, we performed $^{1}H$-$^{15}N$ HSQC NMR titrations. Both compounds show similar chemical

shift perturbations (CSPs) in the $^{1}H$-$^{15}N$ HSQC spectrum (Fig. 3a, Supplementary Fig. 2). The chemical shift changes map to the helices αA and αB, strand β3′ and the loop connecting β1 and helix αA of the SPF45 UHM domain (Fig. 3b, Supplementary Fig. 2). This indicates that the compound binds to the ULM-binding interface of the protein. As 7,8-dimethoxyperphenazine (Cmp7) exhibits affinity similar to the parent compound identified from the HTS (Supplementary Fig. 2), binds to the same site on the SPF45 UHM domain and was found stable in solution, we performed further structural studies with this compound.

First, we tried to obtain intermolecular NOEs between the compound and the SPF45 UHM domain, to obtain direct distance information and to rule out that the chemical shift changes observed are partially allosteric. We recorded 2D and 3D $\omega_1$-isotope-filtered NOE NMR experiments[46] to derive distance restraints based on NOEs between protein residues and

**a**

7,8-Dihydroxyperphenazine
EC$_{50}$ = 10.4 ± 0.7 μM

**b**

Cmp1
EC$_{50}$ = 296.8 ± 109 μM

Cmp2 - Cmp9

**c**

| Cmp | $R_1$ | $R_2$ | $R_3$ | $R_4$ | EC$_{50}$ (μM) |
|------|------|------|------|------|------|
| Cmp2 | H | H | H | | 210.7 ± 72.0 |
| Cmp3 | Cl | H | H | | 213.5 ± 63.1 |
| Cmp4 | Cl | H | H | | 54.4 ± 4.1 |
| Cmp5 | Cl | H | H | | 67.6 ± 8.1 |
| Cmp6 | Cl | OMe | H | | 32.0 ± 8.4 |
| Cmp7 | Cl | OMe | OMe | | 10.9 ± 0.5 |
| Cmp8 | Cl | | OMe | | 7.6 ± 0.3 |
| Cmp9 | Cl | | OMe | | 6.9 ± 0.7 |

**Fig. 2 SAR on 7,8-dihydroxyperphenazine. a** Chemical structure of 7,8-dihydroxyperphenazine is shown. **b, c** SAR studies showing the affinity of each compound for SPF45 as determined by AlphaScreen assay. Errors represent standard error of fitting (*n* = 1).

functional groups of the small molecule. Intermolecular NOEs are observed involving residues in the tryptophan binding pocket of the SPF45 UHM domain and the phenothiazine moiety of the compound. Additional NOEs are observed from Glu329 and Lys332 to the perphenazine tail of the inhibitor (Fig. 3c, d, Supplementary Fig. 3).

In order to determine the binding site and pose of the ligand, we performed HADDOCK-based[47] semi-flexible docking calculations using the CSPs and intermolecular NOE restraints. The four lowest energy structures obtained from these calculations (see Methods) are shown in Supplementary Fig. 3. All four structures show a similar binding pose with the phenothiazine moiety occupying the tryptophan binding pocket of the SPF45 UHM domain (Supplementary Table 1). In the structure (Fig. 3e) Phe377 forms π-stacking interactions with the phenothiazine ring, which fits snugly into the pocket formed by Val382, Phe377, Leu372, Leu309, Met309 and Thr326. NOEs between the perphenazine tail of the compound and Glu329 and Lys332 indicate that the tail binds and extends in a direction opposite to the canonical peptide-binding mode.

**Crystal structure of PUF60 UHM domain-inhibitor complex.** Multiple attempts to obtain crystals of the SPF45 UHM domain in complex with 7,8-dimethoxyperphenazine were not successful. To confirm the binding pose of the compound we pursued crystallization attempts with other UHM domains. This approach is based on the observation that the ULM-binding pockets of UHM domains are highly conserved suggesting that our small molecule inhibitors will bind to other UHM domains using a similar binding mode. To this end, we succeeded in soaking experiments with 7,8-dimethoxyperphenazine and PUF60 UHM crystals formed under conditions reported previously[48] (Supplementary Fig. 4). These crystals diffracted to a resolution of 1.9 Å and unambiguous electron density was found for the compound in two out of four protein molecules in the asymmetric unit (Supplementary Fig. 4). The absence of electron density of the ligand in the other two molecules of the asymmetric unit results from the fact that the binding site is blocked by symmetry-mates in the crystal.

As expected, the phenothiazine moiety of 7,8-dimethoxyper-phenazine occupies the tryptophan-binding pocket (Supplementary Fig. 4). The binding pocket is slightly expanded compared to the recognition of ULM peptide in order to accommodate the three aromatic rings of the phenothiazine. The tricyclic ring system adopts a non-planar bent arrangement due to the presence of the sulfur heteroatom. The hexagonal ring attached to the 7,8-dimethoxy position forms π-stacking interaction with Phe534 of the UHM domain. The chlorine atom snugly fits into the hydrophobic tryptophan-binding pocket of the UHM domain. The 7-methoxy group of the compound points to the pocket that is normally occupied by Arg337 of the ULM peptide ligand, whereas the 8-methoxy group is solvent exposed. Consistent with the NMR data observed for the complex with the SPF45 UHM domain, the piperazine moiety extends in a direction that is opposite to the peptide-binding region (Supplementary Fig. 5). The ion pair of the tertiary N14-amine group of the piperazine moiety interacts with the Glu483 sidechain, which in turn is stabilized via hydrogen bond with the side chain of Lys486 (Fig. 4). This is similarly seen in the structural model of the structure of the SFP45 UHM with phenothiazine (Supplementary Fig. 5). In the SPF45 UHM-SF3b1-ULM peptide complex structure[8], the corresponding Arg375 and Glu329 side chains of the UHM form a salt bridge and parallel stacking interactions with the tryptophan in the ULM peptide (Supplementary Fig. 5). Thus, Arg375 and Glu329 form the tryptophan-binding pocket along with Phe377. However, as the piperazine moiety extends in the direction opposite to the peptide-binding region, the salt-bridge between corresponding positions in PUF60 UHM domain (Arg532 and Glu483) is not present and thus the tryptophan-binding pocket is not completely formed (Supplementary Fig. 5).

**Structure based optimization of 7,8-dimethoxyperphenazine.** The methoxy group at the 7-position of the 7,8-dimethox-yperphenazine ligand in the co-structures of the PUF60 and the SPF45 UHM domains points in a similar direction as the Arg337 side chain in the SF3b1ULM peptide ligand (Supplementary Fig. 5). We therefore substituted the 7-methoxy group

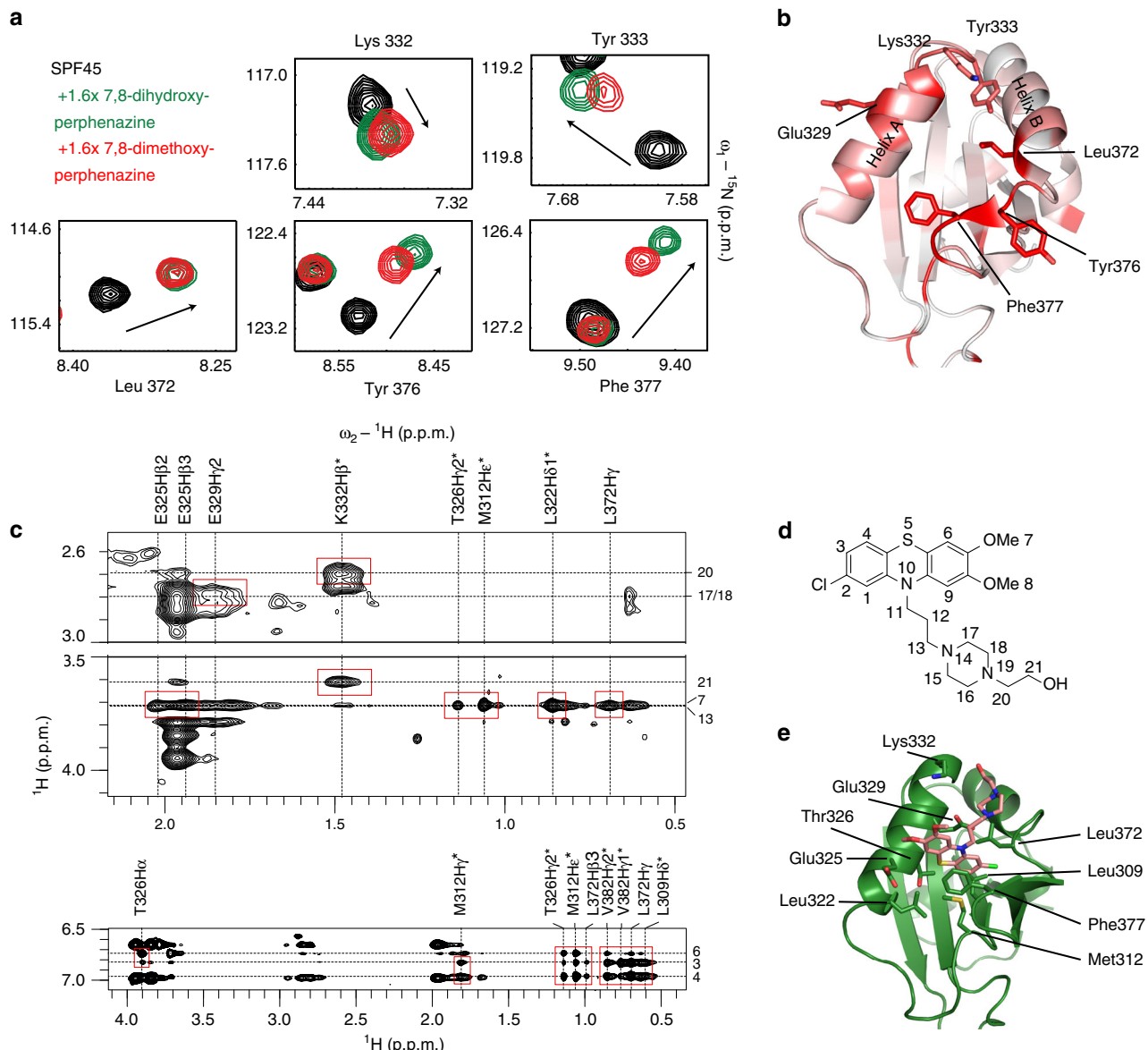

**Fig. 3 Binding site mapping and structural model of the phenothiazine-SPF45 UHM interaction. a** ${}^{1}$H-${}^{15}$N HSQC experiments of 80 μM ${}^{15}$N-labeled SPF45 UHM domain free (black) and in the presence of 1.6-fold molar excess of 7,8-dihydroxyperphenazine (green) or 7,8-dimethoxyperphenazine (red). **b** Residues which show significant amide CSPs upon titration of 7,8-dimethoxyperphenazine are highlighted in red on the structure of the SPF45 UHM domain. **c** 2D ω$_1$-filtered NOEs between the SPF45 UHM domain and 7,8-dimethoxyperphenazine are shown. Red boxes indicate NOEs that were assigned and used in docking calculations. Chemical shift assignments of SPF45 protons and of the compound are shown on top and at the right-hand side of the spectrum, respectively. **d** Chemical structure of 7,8-dimethoxyperphenazine with atom numbering. **e** The lowest energy structure from the HADDOCK cluster is shown with the binding site residues. 7,8-dimethoxyperphenazine is colored in salmon.

with either saturated (Cmp8) or unsaturated (Cmp9) aliphatic chains to enhance direct contacts in this region. This led to a minor affinity increase from 10.9 μM for 7,8-dimethoxyperphenazine to 7.6 μM and 6.9 μM for Cmp8 and Cmp9, respectively (Fig. 2c), consistent with the rationale that aliphatic substituents at the 7-position in 7-alkoxy-8-methoxyperphenazine extend in the direction of the Arg337-binding pocket and mediate hydrophobic contacts in this region.

**In vitro selectivity and splicing inhibition activity of UHM domain inhibitors**. Because the recognition of 7,8-dimethoxyperphenazine, as seen in our structural models with both SPF45 and PUF60 UHMs, involves the highly conserved ULM tryptophan-binding pocket, we expected that the inhibitors will bind to other UHM domains as well. To assess the selectivity of

these inhibitors, we compared the affinities of these inhibitors for three different UHM domains of SPF45, PUF60 and U2AF65 (Table 1). All three compounds (Cmp7-9) showed similar affinity for the three different UHMs confirming that the inhibitors are mainly recognized by the conserved tryptophan binding pocket of the UHM domains and there are no significant specific contacts made. Considering that several other UHM domains are known, these results imply that our inhibitors will possibly bind to all UHM domains and would represent general UHM inhibitors.

In order to assess the functional activity of the phenothiazine compounds, we performed in vitro splicing assays. Initially, we tested the effect of 7,8-dimethoxyperphenazine (Cmp7) on the splicing of pre-mRNAs with splice sites with more (MINX pre-mRNA) or less (IgM pre-mRNA) pronounced pyrimidine tracts in HeLa nuclear extract. The inhibition efficiency with Cmp7 was

slightly better with the IgM pre-mRNA, where A complex formation was essentially completely abolished at 2 mM while with MINX very low levels of the A complex was still observed at 2 mM (Supplementary Fig. 6). Therefore, for further splicing assays, IgM pre-mRNA was chosen as substrate as it harbors 3′ splice sites of intermediate strength and is thus expected to be more sensitive to regulation. Similar to Cmp7, Cmp9 abolished the catalytic steps of splicing at 2 mM concentration whereas Cmp8 inhibited splicing completely at 1.5 mM (Fig. 5a). The fact that the splicing inhibition activity is significantly weaker compared to the binding affinity of the inhibitors in vitro suggests that the compounds may bind to other factors in the nuclear extracts. As splicing inhibition was first observed at millimolar concentrations of the inhibitors, we cannot completely rule out that we induce inhibitory effects not related to UHM binding. To assay at which stage the inhibitors block splicing, we analyzed splicing complex formation on a 2% agarose gel. All three compounds inhibited the formation of spliceosomal complex A at those concentrations where splicing was completely abolished; with Cmp7 and Cmp9, only very low levels of the A complex appeared to still form (Fig. 5b). As mentioned previously, during the complex E/H to complex A transition, the interaction of the SF1 ULM with the U2AF65 UHM domain in complex E is replaced by SF3b1 ULM interactions in the

complex A. Our data are thus consistent with the idea that the compounds indeed inhibit UHM/ULM interactions during early spliceosome assembly in HeLa nuclear extracts.

We also tested the splicing inhibition activity of 7,8-dihydroxyperphenazine, the parent compound discovered in the HTS (Fig. 6). Surprisingly, this compound completely inhibited splicing of MINX pre-mRNA at 50 μM concentration and led to an accumulation of spliceosomal A complexes, in which the U1 and U2 snRNPs are bound to the 5′ splice site (5′ ss) and the so-called branch site just upstream from the 3′ ss of the intron, respectively (Fig. 6a, b). Thus, 7,8-dihydroxyperphenazine inhibits the conversion of an A complex into a B complex during spliceosome assembly. The transition from complex A to B involves two steps: initial docking of the U4/U6.U5 tri-snRNP to the A complex to form pre-B complexes (which do not withstand analysis on our agarose gels) and a subsequent rearrangement that leads to its stable integration (i.e., complex B formation). As pre-B complex formation is difficult to assay for with a standard pre-mRNA substrate, where splicing complex formation occurs across the intron, we switched to a cross exon system, using the MINX exon RNA that contains an exon flanked upstream by a branch-site/3′ss and downstream by a 5′ss. 7,8-dihydroxyperphenazine had no effect on the formation of cross-exon A-like complexes containing U1 and U2, but the formation of a B-like complex containing stably associated tri-snRNP was not observed in its presence upon addition of a 5′ss RNA oligonucleotide that mimics an upstream, cross-intron 5′ss, as assayed by agarose gel electrophoresis and also gradient centrifugation (Fig. 6c, d). Glycerol gradient centrifugation of cross-exon complexes showed that the inhibitor leads to the formation of a complex with a sedimentation coefficient lower than that of the 37S exon complex formed in the absence of inhibitor. The 37S complex contains U1 and U2 plus loosely associated U4/U6.U5 tri-snRNP[49], suggesting that the compound inhibits U4/U6.U5 tri-snRNP docking onto the A-like complex (Fig. 6d, e). Indeed, affinity purification revealed that very little tri-snRNP is docked onto the A-like complexes in the presence of inhibitor, as evidenced by the reduced levels of U4, U5, and U6 snRNA in the purified, inhibited complexes (Fig. 6e).

## Discussion
The dynamic nature of the spliceosome, which involves the assembly and rearrangement of multiple splicing factors at the pre-mRNA splice sites, poses a significant challenge for detailed mechanistic studies by biochemical and structural methods. Although major breakthroughs have been obtained for structural studies of many stages of the spliceosome cycle using cryo-

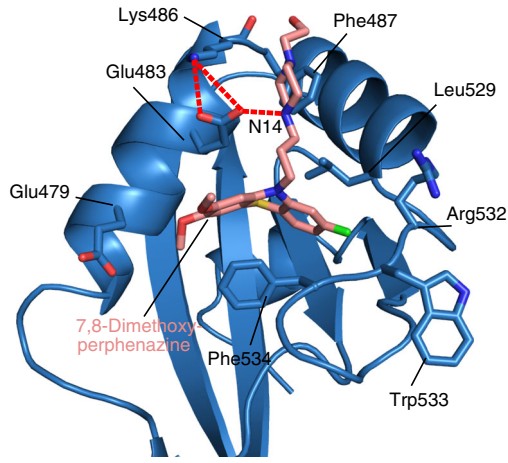

**Fig. 4 Crystal structure of PUF60 UHM in complex with 7,8-dimethoxyperphenazine.** Side chains are shown for residues in the binding site. 7,8-dimethoxyperphenazine is colored in salmon. A polar contact between Glu483 and the N14-amine group of the piperazine moiety is shown by a red dotted line.

**Table 1 IC$_{50}$ values for the binding of phenothiazine derivatives to SPF45, PUF60 and U2AF65 UHM domains as determined by AlphaScreen assay.**

| Compound | SPF45 UHM IC$_{50}$ (μM)* | PUF60 UHM IC$_{50}$ (μM)* | U2AF65 UHM IC$_{50}$ (μM)* |
|---|---|---|---|
| 7,8-dihydroxyperphenazine | 10.4 ± 0.7 | 10.4 ± 0.5 | 9.2 ± 0.9 |
| Cmp1 | 296.8 ± 109 | 178.1 ± 47.5 | 305.3 ± 133 |
| Cmp2 | 210.7 ± 72.5 | 84.2 ± 10.6 | 135.5 ± 35.3 |
| Cmp3 | 213.5 ± 63.5 | 96.0 ± 14.0 | 268.0 ± 108 |
| Cmp4 | 54.5 ± 4.1 | 47.8 ± 3.0 | 49.2 ± 4.4 |
| Cmp5 | 67.6 ± 8.1 | 61.8 ± 4.9 | 67.6 ± 7.7 |
| Cmp6 | 32.0 ± 1.7 | 42.5 ± 9.6 | 45.9 ± 9.5 |
| Cmp7 | 10.9 ± 0.5 | 11.7 ± 0.4 | 12.1 ± 0.5 |
| Cmp8 | 7.6 ± 0.3 | 7.2 ± 0.3 | 7.0 ± 0.3 |
| Cmp9 | 6.9 ± 0.7 | 6.6 ± 0.5 | 7.0 ± 0.8 |

*Errors represent standard errors of fitting.

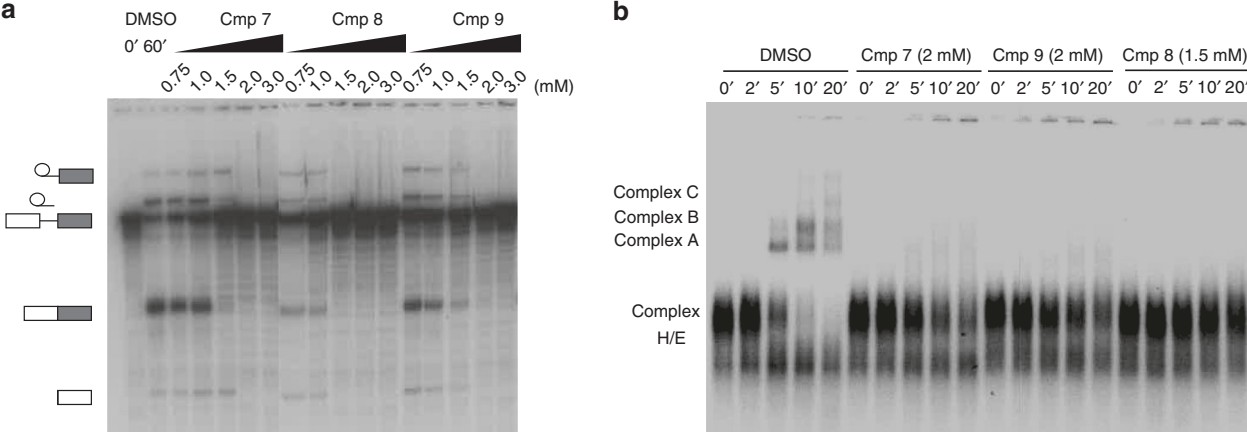

**Fig. 5 In vitro splicing assay of IgM pre-mRNA with 7,8-dimethoxyperphenazine (Cmp7), Cmp8 and Cmp9. a** 7,8-dimethoxyperphenazine (Cmp7), Cmp8 and Cmp9 abolish splicing of IgM pre-mRNA at 2 mM, 1.5 mM and 2 mM respectively. **b** Separation of spliceosomal complexes on an agarose gel. For Cmp7, splicing inhibition (**a** and **b**) was replicated in three independent experiments, and for Cmp8 and Cmp9, it was performed once. The uncropped gel images are included in the Source Data file.

**Fig. 6 The effects of 7,8-dihydroxyperphenazine on in vitro splicing and cross-exon complex formation. a** 7,8-dihydroxyperphenazine completely inhibits MINX pre-mRNA splicing at 50 μM and (**b**) inhibits spliceosome assembly after complex A formation. **c–e** 7,8-dihydroxyperphenazine does not affect cross-exon A-like formation but inhibits U4/U6. U5 tri-snRNP docking to the cross-exon A-like complex. Anacardic acid (AA), which disrupts the U4/U6.U5 tri-snRNP, was used as a positive control to stall assembly at the A-like complex stage. Pre-mRNA splicing inhibition experiments, and gradient centrifugation and affinity selection of the cross-exon complexes were performed in at least two independent experiments with similar results. The Uncropped gel images are included in the Source Data file.

EM[50,51], our understanding of structural arrangements in the early complex E is still limited. Small molecule inhibitors, which can specifically target interactions during early spliceosome assembly, are useful tools to stall and stabilize such complexes to probe their topology, structure and function. Most splicing inhibitors discovered in recent years have been identified by HTS with in vitro or cell-based assays, and the molecular targets of these compounds within the spliceosome are in many cases unknown. Numerous diseases are associated with aberrant splicing and on-going developments of therapies rely on a modulation of pre-mRNA splicing[3,38,39]. In this regard, targeting essential protein-protein interactions at early stages of spliceosome assembly may provide new therapeutic opportunities.

Here, we have identified phenothiazines as inhibitors of UHM-ULM interactions. Our NMR and crystallographic structural data show that phenothiazines target the tryptophan binding pocket of UHM domains. SAR obtained with 7,8-dihydroxyperphenazine derivatives identified moieties required for the interaction of this compound with the UHM domains. Removal of the chlorine and hydroxyl substituents of the 7,8-dihydroxyperphenazine and modification of the tricyclic phenothiazine core led to an almost complete loss of the binding affinity of these compounds, while the two hydroxyl groups at the 7- and 8-positions were successfully replaced with methoxy groups without affecting the interaction.

We have also shown, as a proof-of-concept, that our inhibitors indeed are active and impair early spliceosome assembly in splicing assays. 7,8-dimethoxyperphenazine, Cmp8 and Cmp9 inhibit complex A formation, consistent with an inhibition of UHM-ULM interactions. Several UHM-ULM interactions are implicated in the formation of spliceosomal E and A complexes. For complex E formation, UHM-ULM interactions between the U2AF65 UHM and the SF1 ULM and the U2AF35 UHM and the U2AF65 ULM are required. During complex E to complex A transition, the SF1 ULM is replaced by SF3b1ULMs that bind to the U2AF65 UHM. Therefore, it is plausible that the inhibitors target both U2AF65 and U2AF35 UHMs. Note, that a number of additional UHM-containing splicing factors and paralogs of U2AF65 and U2AF35 are implicated in AS regulation of specific genes that may affect early spliceosome assembly, which could be additional targets of our inhibitors (Supplementary Fig. 1).

Unexpectedly, the parent compound 7,8-dihydroxyperphenazine blocks formation of complex B. This apparently is caused by inhibiting the initial, less stable docking of the U4/U6.U5 tri-snRNP to the A complex during formation of the pre-B complex. The observation that 7,8-dihydroxyperphenazine inhibits a different stage of spliceosome assembly (pre-B vs A complex) and at significantly lower concentrations (50 µM) (Fig. 6) compared to 7,8-dimethoxyperphenazine (2 mM) in our assays, strongly argues that different molecular mechanisms and targets are involved. One explanation for the distinct activity could be that the simpler decoration of the 7,8-dihydroxyperphenazine enables higher affinity binding to factors involved in complex B formation, while the bulkier derivatives may not be able to mediate similar interactions due to steric constraints. Another possibility may be linked to the known observation that 7,8-dihydroxyperphenazine undergoes oxidative degradation and thus could potentially generate reactive oxygen species (ROS)[44]. In fact, it was previously shown that quinone-containing compounds that are capable of generating ROS can inhibit spliceosome assembly[36]. This activity was attributed not simply to their binding to spliceosome components but by the ability to trigger oxidation of labile cysteines residues of proteins required for spliceosome assembly. It is thus a possibility that 7,8-dihydroxyperphenazine blocks spliceosome assembly by oxidizing labile cysteines of proteins specifically found in spliceosomal A

complexes, or in the U4/U6.U5 tri-snRNP, inhibiting its docking. In addition to this, the weak splicing inhibition activity of Cmp7-Cmp9 compared to its in vitro binding affinity to the UHM domains suggests the binding of these compounds to other factors in the nuclear extract. A similar effect was also observed previously for our cyclic peptide inhibitors, which could very specifically target U2AF65 UHM domain with low nM affinity in vitro[43].

In summary, we report a proof-of-concept study demonstrating that phenothiazines can bind to UHM domains and thereby modulate spliceosome assembly in vitro. Initial SAR and distinct activities of phenothiazine derivatives indicate the potential for the development of UHM inhibitors with improved affinity and specificity, which may be useful for probing the structure and regulation of early spliceosome assembly.

## Methods

**Protein expression and purification**. His-tagged UHM domains of SPF45 (residues 301–401)[8], U2AF65 (residues 372–475)[19] or PUF60 (residues 460–559)[20] were expressed in *Escherichia coli* BL21 (DE3) cells in either LB media or minimal media supplemented with 0.5 g/L $^{15}NH_4Cl$ (for $^{15}N$ labelling) or 0.5 g/L $^{15}NH_4Cl$ and 2 g/L $^{13}C$ glucose (for $^{15}N$, $^{13}C$ labelling). Protein expression was induced with 0.5 mM IPTG at 20 °C overnight. Proteins were purified by His-tag affinity chromatography using Ni-NTA resin. In case of protein used for NMR and ITC, affinity chromatography was further followed by overnight TEV protease cleavage to cleave His tag, 2nd Ni-column to remove TEV protease and His tag and size exclusion chromatography. In case of protein used for AlphaScreen assay and PUF60 crystallization, affinity chromatography was directly followed by size exclusion chromatography. Purified proteins were concentrated and stored at −80 °C till further use.

**NMR experiments**. For SPF45 UHM-inhibitor NMR titrations, $^1H$, $^{15}N$ heteronuclear single quantum correlation (HSQC) experiments were acquired at 298 K on Bruker Avance III spectrometers at proton Larmor frequencies of 600 MHz, 750 MHz or 800 MHz equipped with triple resonance cryogenic (600 and 800 MHz) or room temperature (750 MHz) gradient probes. 80 µM of $^{15}N$-labelled SPF45 UHM domain in buffer containing 20 mM potassium phosphate pH 6.8, 150 mM NaCl, 5 mM DTT supplemented with 10% $D_2O$ was titrated with various concentrations of inhibitor. Spectra were acquired using TopSpin 3.2, processed with NMRPipe/Draw[52] and analyzed using CCPN Analysis[53]. Backbone chemical shift assignments of the SPF45 UHM domain were used (BMRB ID: 15882) together with $^1H$, $^{15}N$-HSQC titrations to obtain assignments of the protein-7,8-dimethoxyperphenazine bound protein. The residues showing CSPs one-standard deviation above average CSP values were fitted by nonlinear regression analysis in GraphPad Prism 6 to obtain a group dissociation constant ($K_D$). For assignment of signals in the $\omega_1$-filtered NOESY experiments[54], amino acid side chains which show NOEs to Cmp7 were assigned using 3D H(CCO)NH and CC(CO)NH experiments recorded on a sample containing 650 µM $^{15}N$, $^{13}C$ labeled SPF45 and 2-fold excess of 7,8-dimethoxyperphenazine to completely saturate the protein binding site and thus avoid NMR signals of the free protein. 2D $\omega_1$-filtered NOESY and 3D $\omega_1$-filtered, edited $^{13}C$ NOESY experiments were recorded in 100 % $D_2O$ to obtain inter-molecular NOEs between the protein-7,8-dimethoxyperphenazine complex.

**Structural modelling**. For modeling the SPF45 UHM-Cmp7 complex, HADDOCK webserver[47] was used. Hydrogens were added to the crystal structure of SPF45 (PDB ID: 2PEH) using Molprobity[55] while the 7,8-dimethoxyperphenazine input file was created using the PRODRG webserver[56]. Residues showing highest CSPs in SPF45 were used to derive ambiguous interaction restraints (Met 312, Leu322, Thr326, Glu329, Lys332, Tyr333, Asp371, Leu372, Arg375, Phe377 and Val382). The 32 inter-molecular NOEs obtained from the NMR $\omega_1$-filtered experiments were included as unambiguous distance restraints using standard HADDOCK settings. Subsequent water refinement yielded a single cluster with 50 models. Detailed structural statistics of the HADDOCK-derived cluster are provided in Supplementary Table 1.

**PUF60-small molecules crystallization and data processing**. Crystals were obtained by vapor diffusion hanging drop method by mixing 1 µl protein (at 75 mg/ml concentration in 20 mM Tris pH 7.0, 50 mM NaCl, 1 mM BME) and 1 µl of well solution (1.4 M $(NH_4)_2SO_4$ and 50 mM K-formate) as published before[48]. Crystals suitable for diffraction grew in 4–5 days, which were soaked overnight in a 2 µl of fresh solution containing 1.5 M $(NH_4)_2SO_4$, 50 mM K-formate and 1 mM 7,8-dimethoxyperphenazine (suspended in 1 µl solution of 1.5 M $(NH_4)_2SO_4$, 50 mM K-formate) overnight. Crystals were cryoprotected by serial transfer into a solution of 1.5 M $(NH_4)_2SO_4$, 50 mM K-formate, 1 mM 7,8-dimethoxyperphenazine and 20% ethylene glycol. Diffraction datasets were collected at

beamlines of the European synchrotron research facility (ESRF), Grenoble, France. Datasets were integrated and scaled with the XDS[57] software package. All structures were solved with molecular replacement using the crystal structure of thioredoxin-tagged PUF60 (PDB id: 3DXB) as the search model in Phaser[58]. The inhibitors and the missing residues were built in the visible electron density using the Coot[59] model building software. Coordinates and the restraints files for the inhibitors were obtained from the PRODRG server[56]. The built models were refined with in Phenix suite[60]. Figures were made using The PyMOL molecular graphics System.

**Isothermal titration calorimetry (ITC)**. Prior to ITC experiments, the SPF45 UHM domain was dialyzed overnight into the ITC buffer containing 20 mM potassium phosphate, 150 mM NaCl and 1 mM β-mercaptoethanol, pH 6.8. ITC experiments were performed at 25 °C by titrating peptide into 50 μM of SPF45 UHM domain. 500 μM of cyclic peptide was used in the syringe and 1.5 μl of volume was titrated into the cell. The experiments were performed with ITC200 Microcal system and the data were fitted with the Origin software provided with the instrument using a one-site binding model.

**Fluorescence polarization assay**. Fluorescence polarization (FP) assays were carried out in 384-well plate. The buffer conditions used are 20 mM Tris pH 7.5, 300 mM NaCl, 1 mM DTT, 100 nM tracer and 1.6 % DMSO optimized for maximal signal gained in multiple rounds of assay optimization. During assay optimization, several pHs of the buffer and concentrations of NaCl and tracer were tested. For binding curves, a constant amount of tracer (100 nM) was titrated into serial dilutions of protein. The assay was carried out in 40 μl volume and the protein tracer mixture was incubated for 1 h before reading the plate. The polarization was measured in an Envision plate reader (Perkin Elmer, Waltham, MA) using FP480 (excitation) and FP533 (emission filters). The data were plotted against the log protein concentration and was fitted with OriginPro software using a 4-parameter logistic nonlinear regression model to obtain the binding curve. The binding curve was used to define the $EC_{80}$ concentration (concentration of protein required to achieve 80% of maximal response units) which was further used in the high-throughput screening (HTS).

**High-throughput screening**. A HTS by means of single point titration in the FP assay was carried out. The protein (28 μl of 1.4× $EC_{80}$ concentration calculated from the binding curve) was added to the assay plate by the MultidropCombi reagent dispenser and the buffer tracer mix (12 μl of 333.3 nM tracer + 5.3% DMSO stock solution in FP buffer) was added by the robot. The protein concentration used in the HTS was taken as that required for achieving 80% of maximum response in FP units for tracer binding. It was determined from the binding curve obtained by titrating a series of protein concentrations in 100 nM of tracer. The protein concentration used in the assay varied from batch to batch (of purification) and was determined anew for each batch. The last two columns of the assay plate were used for controls in each plate. The controls in the plates included DMSO control (negative control with no compound), positive control (100 μM of peptide 10 cyclic peptide)[43], high protein and no protein controls (to obtain the assay window). The compound library was also provided in 384-well format. Compounds were pipetted directly into the assay plate to a final concentration of 125 μM. After addition of the compounds, the plates were sealed and incubated at 25 °C for 1 h in the dark.

**AlphaScreen assay**. AlphaScreen assay was carried out using streptavidin donor beads to bind biotinylated peptide and $Ni^{2+}$ acceptor beads to bind His-tagged protein. The assays were carried out by incubating 10 nM protein with 10 nM biotinylated peptide (biotinylated-RKSRWDETP) in 20 mM Potassium phosphate, 150 mM NaCl, 0.5% bovine serum albumin, 0.05% NP40 detergent, 1 mM BME and 1% DMSO for 30 min. In case of competition assay, the desired small molecule inhibitor was added and the mixture was incubated in dark for 30 min. The data were plotted in OriginPro 9.0 software using nonlinear dose response curve fitting function.

**In Vitro splicing and cross-exon complex formation**. HeLa nuclear extract was prepared as previously described[61]. Uniformly $^{32}$P-labelled MINX or IgM pre-mRNA, or MINX exon RNA containing MS2-aptamers at its 3′ end, was generated by in vitro transcription. Splicing reactions contained 50% (v/v) HeLa nuclear extract, 50 mM KCl, 3 mM $MgCl_2$, 2 mM ATP, 20 mM creatine phosphate and 10 nM uniformly $^{32}$P-labelled, $m^7$G-capped pre-mRNA or MINX Exon RNA, plus the indicated amount of inhibitor and were incubated at 30 °C for the indicated times. To isolate cross-exon complexes lacking U4/U6.U5 tri-snRNPs, 0.5 mM anacardic acid (AA), which disrupts the tri-snRNP, was added to the reaction[62]. For pre-mRNA splicing, RNA was recovered and separated on a 14% denaturing polyacrylamide gel. Unspliced pre-mRNA, as well as the splicing intermediates and products, were detected using a Typhoon phosphoimager (GE Healthcare). Spliceosomal complex formation (cross-intron or cross-exon) was analyzed by electrophoresis in a 2% (w/v) low melting agarose gel in TBE buffer (89 mM Tris-HCl, pH 7.5, 89 mM boric acid, 2.5 mM EDTA, pH 8.0) in the presence of 0.5 μg/μl heparin, and bands were visualized with a Typhoon phosphoimager (GE

Healthcare). To convert a cross-exon complex into a B-like complex with stably associated U4/U6.U5 tri-snRNP, cross-exon complexes were first allowed to form for 3 min and subsequently a 100-fold molar excess of a 5′ss RNA oligonucleotide, 5′-AAGGUAAGUAU-3′, which mimics an upstream cross-intron 5′ss, was added and the reaction was incubated for an additional 3 min (gradient analysis) or 5–30 min (agarose gel analysis). Splicing reactions were loaded onto a linear 10–30% (v/v) glycerol gradient containing G-150 buffer (20 mM HEPES-KOH pH 7.9, 1.5 mM $MgCl_2$, 150 mM KCl) and centrifuged at 60000 rpm for 135 min at 4 °C in a Sorvall TH660 rotor. The gradients were harvested manually in 175 μl fractions from the top and the distribution of $^{32}$P-labeled MINX Exon RNA was determined by Cherenkov counting. Cross-exon complexes in the peak gradient fractions were purified by MS2 affinity selection as described previously[49]. RNA was recovered from the purified complexes, separated by denaturing PAGE and visualized by silver staining.

**Chemical synthesis**. Synthesis of the tracer peptide used for the for high-throughput screening and phenothiazine derivatives are described in the Supplementary Methods.

**Reporting summary**. Further information on research design is available in the Nature Research Reporting Summary linked to this article.

## Data availability
Coordinates and structure factors for the PUF60 UHM domain in complex with 7,8 dimethoxyperphenazine have been deposited in the PDB with accession number 6SLO [https://doi.org/10.2210/pdb6SLO/pdb]. NMR restraints for the SPF45 UHM complex with 7,8-dimethoxyperphenazine are deposited in the BMRB under the accession code 50299. Other data are available from the corresponding authors upon reasonable request. Source data are provided with this paper.

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

## Acknowledgements

P.K.A.J acknowledges support by a Boehringer Ingelheim Fonds doctoral fellowship. T.K. and H.K.P. acknowledge funding by the Lifescience Stiftung and the Helmholtz Drug Discovery Initiative. K.S. is grateful for support by the IMPRS-LS graduate school. We acknowledge the NIMH chemical synthesis and drug supply program for the supply of 7,8 dihydroxyperphenazine dihydrochloride. We thank Gabi Heyne (MPI Göttingen) for excellent technical assistance, and Thorsten Berg, University of Leipzig for discussions during early stages of the project. We acknowledge access to the Bavarian NMR Center (Garching) (www.bnmrz.org) and the NMR facility of the European Molecular Biology Laboratory (Heidelberg) for NMR measurement time. This work was supported by the DFG (grants SFB1035, GRK1721 to M.S and SFB806, project A1 to R.L).

## Author contributions

P.K.A.J. performed NMR titrations, crystallographic structure determination, and bio-physical characterization. T.G.K., P.K.A.J., D.G. and H.K. designed the cyclic peptide, T.G.K. performed its chemical synthesis. T.K., and H.K.P. performed chemical synthesis of small molecules. D.G, P.K.A.J., and F.H. designed and optimized the FP assay and P.K.A.J., C.S. and F.H. performed HTS screening. K.So. performed NMR experiments and structural modelling. P.K.A.J., K.Sch. and K.H. optimized and performed AlphaScreen assays. C.L.W. performed in vitro splicing assays and C.L.W. and R.L. analyzed splicing assays. D.G., P.K.A.J, M.S. and T.K. designed the study. P.K.A.J., T.K., T.B. and M.S. wrote the paper. All authors commented and approved the final version of the paper.

## Funding

## Competing interests

The authors declare no competing interests.
