## [Peer Review File · Nature Communications]

REVIEWER COMMENTS

Reviewer #1 (Remarks to the Author):

ULM-UHM interactions play a major role in the early steps of the spliceosome assembly. To date there are no known compounds which allow us to examine these interactions during spliceosome assembly. This paper describes the identification and characterisation of phenothiazine derivatives as UHM domain inhibitors and their ability to inhibit pre-mRNA splicing in vitro. Using a fluorescence polarization assay a library of 43296 small molecules were screened for disruption of SF3B1 ULM5 - SPF45 UHM binding. This HTS screen and additional secondary assays identified 7,8-dihydroxyperphenazine as a hit compound. Structure activity studies lead to the identification of more stable analogues and highlighted the importance of the phenothiazine ring for the interaction between compound and UHM domain. A detailed structural analysis of this interaction identified binding sites and how the phenothiazine derivatives fit into the tryptophan pocket of the UHM domain. Affinity studies show that 7,8-dihydroxyperphenazine and its analogues have roughly the same binding affinity to three different UHM domains (SPF45, PUF60 and U2AF65), indicating that these molecules are general UHM inhibitors. In order to demonstrate that phenothiazine derivatives do interfere with the splicing process, 7,8-dihydroxyperphenazine as well as compound 7, 8, 9 were tested in in vitro splicing reactions. All four compounds inhibited pre-mRNA splicing in vitro. Compound 7, 8, 9 stalled the spliceosome assembly at the E complex formation whereas the parent compound was shown to block the docking of the tri-snRNP to the A-like complex using a cross exon system. In addition, the parent compound is 40x more potent than the other compounds. These differences are probably due to oxidative degradation of 7,8-dihydroxyperphenazine and raises the possibility that the parent compound inhibits pre-mRNA splicing through alternative mechanisms and targets and not through UHM inhibition.

The study of the splicing process has been hampered by the very limited number of well characterised compounds that specifically interfere with the splicing process. Therefore, the identification and characterization of these novel splicing inhibitors will greatly benefit our understanding of the spliceosome assembly.

However, there are some issues that I think should be clarified/addressed.

Major flaws:

- 1.) The authors state in the introduction that one of the main reasons for conducting this HTS screen was to identify small molecules that inhibit the UHM domain, because trafficking peptides across the cell membrane is challenging. However, the authors do not return to this point in the article.
 - Have these compounds be tested in cells? If not, could these compounds be used for studying the splicing process in cells, given that these compounds inhibit most likely all UHM domains?
 - How do these compounds compare to the UHM peptide inhibitors in in vitro splicing reactions?
- 2.) A diagram showing all known UHM-ULM interactions formed during spliceosome assembly and the proposed consequences for spliceosome assembly if these UHM-ULM interactions are inhibited would be useful for the reader.
- 3.) The IgM pre-mRNA was used for testing compounds 7 - 9 in in vitro splicing reactions. The parent compound however was tested using the MINX pre-mRNA. The type of pre-mRNAs used in in vitro splicing reactions can influence the efficiency as well as the effect of a compound on in the splicing reactions. To allow a better comparison between the compounds the same pre-mRNA, either IgM or MINX, should be used.
- 4.) It is most likely that the additional bands in the splicing reactions containing compound in

Figure 6A (running between the MINX pre-mRNA and MINX RNA) are already present in the input RNA. To exclude the possibility that these bands are a result of the treatment, a lane showing the input pre-RNA would be useful.

5.) In the discussion, the authors argue that one reason for 7,8-dihydroxyperphenazine behaving differently to compound 7-9 in splicing reactions, is that "7,8-dihydroxyperphenazine binds to UHM domains involved in splicing complex formation and thereby may trigger the oxidation of labile cysteines in spliceosome associated proteins".

- Would that not be an argument would that not be an argument for the spliceosome to be stalled at the E-complex formation?

- Are there any known UHM-ULM interaction involved in the transition from the A to the B complex?

Minor flaws:

1.) Line 78/79 "Only few inhibitors are known to act during the early phase of spliceosome assembly, i.e. the formation of the early E complex...."

- Which inhibitors are known to stall the spliceosome assembly at the E complex formation?

2.) Line 99: "loss in binding affinity of the probe to the UHM domain"

- SPF45 UHM domain?

3.) Line 107: Figure legend 1 "the binding affinity of this interaction is"

- The binding affinity of the SPF3B1 ULM - SPF45 UHM interaction?

4.) Line 113/114: "at a fixed concentration"

- What concentration?

5.) In Figure 6 C the authors perform in vitro splicing reactions using "the MINX exon RNA that contains an exon flanked upstream by a branch-site/3'ss and downstream by a 5'ss". The splicing reactions are carried out in the presence and absence of 5'ss oligonucleotides. Are these antisense oligonucleotides to block the 5'ss? These oligos are not mentioned in the material and methods section.

6.) In Figure 6 D and E anacardic acid was used to inhibit tri-snRNPs binding to the A like complex, this however is not mentioned in the text of the results section, only in material and methods.

7.) Supplementary figure 2 legend: SPF45 UHM domain?

8.) Supplementary figure 3 legend: SPF45 UHM domain?

Reviewer #2 (Remarks to the Author):

This manuscript describes a successful HTS campaign to discover novel inhibitors of the UHM-ULM interaction, which is a central feature of complexes that regulate splicing. The results are compelling, the compounds are important new tools for dissecting splicing, and the work represents a key first step toward generating bioactive inhibitors of splicing as therapeutics.

Minor comments:

- 1). How does the Haddock pose compare to the co-crystal structure? This could be included in SI.
- 2). It is interesting that the more potent compounds show less selectivity among UHM domains. This could be a potential challenge as these domains have conserved structure due to interactions with multiple ULMs. Are there any "exo-sites" that could be used to build in additional selectivity?
- 3). While it is impressive that the compounds are active in splicing extracts, their IC-50 are nearly 3 orders of magnitude larger than the Kd for binding. The authors suggest an abundant reservoir of other binding sites sequestering the compound. There is a danger that the splicing inhibition is due to millimolar inhibitor inducing effects not related to UHM binding. This might be difficult to untangle experimentally, but could be mentioned.
- 4). On this point, it is stated that the inhibitors cannot compete against the U2AF35-U2AF2-65 complex due to the nanomolar affinity. Perhaps, but this would require some knowledge of the concentrations of those components, since two competitors will compete equally if they are present at their respective Kds. Mass action can overcome even a nM interaction.
- 5). Presumably, the compounds have been tested in cells...

Overall, this is an impressive study, carefully done with a wide range of techniques applied. These compounds will be powerful tools too build on in probing the mechanism of splicing regulation. Also, it is important to note that protein-protein interactions are notoriously difficult to inhibit with small molecules, and this work is a splendid example of this that can serve as a paradigm.

Reviewer #3 (Remarks to the Author):

The manuscript by Jagtap and co-workers discuss the identification of phenothiazines as general inhibitors of the interaction between UHM domains and the cognate ULM peptides, a type of interaction which is key to the assembly of splicing complexes. The authors use a range of techniques, including fluorescence polarization, isothermal titration calorimetry, AlphaScreen assays, NMR, crystallography, molecular modelling and splicing assays to identify phenothiazines as inhibitors of the interaction, characterize the binding and inhibitor activities of different compound of this class and design optimized compounds. They conclude phenothiazines have the potential to become important tools for investigating splicing and alternative splicing, and may be of therapeutic relevance. The manuscript is clearly written and well organized. The Figures are informative and the conclusions are interesting and overall well supported.

A few points should be addressed prior to publication:

- 1) One important question is whether these compounds can target specific UHM-ULM interactions or represent general inhibitors. In vitro, similar EC50 is observed for the different compounds. However, the authors observe a specificity of action in the splicing assays performed. They discuss two possible causes. Clearly, it is very difficult to give a full account of the actions on the compounds on splicing at this stage, but the paper would very much benefit if the two possibilities were explored.
- 2) The authors mention "...some inhibitors during the early phase of spliceosome..." in lines 78-80. This work must be referenced, this information is important to place the results in the context of the literature.
- 3) Does the cyclic peptide discussed in lines 96 to 101 bind like an ULM peptide? The authors refer to previous work - could they elaborate on this statement detailing the evidence presented in that paper? Is additional experimental evidence, e.g. NMR, required here? It is unclear at the moment.

4) In the Methods the authors mention they titrate the two compounds into the protein and record NMR experiments. Did they only use a 1:1 ration? It would be interesting to know ratios they used for the titration and show the spectra in the Supplementary Materials? Also, it would be interesting to see a binding curve.

5) Again in the Methods, the authors mention a 2:1 ratio is used to record two of the NMR experiments. Has this ratio be used for other experiments? Maybe the authors can spell out the rationale of this choice – some of the readers less experienced in this type of studies may wonder.

6) Line 185. Crystal structure of PUF60 UHM domain-inhibitor complex in bold.

7) Line 219. Structure-based optimization of 7,8-dimethoxyperphenazine in bold.

8) Replace supplementary table s1 by Supplementary Table S1 (line 373).

9) Change the font in lines 449, 450, 451, 455, 463, and 471.

Point-by-point response

Reviewer #1:

ULM-UHM interactions play a major role in the early steps of the spliceosome assembly. To date there are no known compounds which allow us to examine these interactions during spliceosome assembly. This paper describes the identification and characterization of phenothiazine derivatives as UHM domain inhibitors and their ability to inhibit pre-mRNA splicing in vitro. Using a fluorescence polarization assay a library of 43296 small molecules were screened for disruption of SF3B1 ULM5 - SPF45 UHM binding. This HTS screen and additional secondary assays identified 7,8-dihydroxyperphenazine as a hit compound. Structure activity studies lead to the identification of more stable analogues and highlighted the importance of the phenothiazine ring for the interaction between compound and UHM domain. A detailed structural analysis of this interaction identified binding sites and how the phenothiazine derivatives fit into the tryptophan pocket of the UHM domain. Affinity studies show that

7,8-dihydroxyperphenazine and its analogues have roughly the same binding affinity to three different UHM domains (SPF45, PUF60 and U2AF65), indicating that these molecules are general UHM inhibitors. In order to demonstrate that phenothiazine derivatives do interfere with the splicing process, 7,8-dihydroxyperphenazine as well as compound 7, 8, 9 were tested in in vitro splicing reactions. All four compounds inhibited pre-mRNA splicing in vitro. Compound 7, 8, 9 stalled the spliceosome assembly at the E complex formation whereas the parent compound was shown to block the docking of the tri-snRNP to the A-like complex using a cross exon system. In addition, the parent compound is 40x more potent than the other compounds. These differences are probably due to oxidative degradation of 7,8-dihydroxyperphenazine and raises the possibility that the parent compound inhibits pre-mRNA splicing through alternative mechanisms and targets and not through UHM inhibition.

The study of the splicing process has been hampered by the very limited number of well characterised compounds that specifically interfere with the splicing process. Therefore, the identification and characterization of these novel splicing inhibitors will greatly benefit our understanding of the spliceosome assembly.

However, there are some issues that I think should be clarified/addressed.

Major flaws:

1.) The authors state in the introduction that one of the main reasons for conducting this HTS screen was to identify small molecules that inhibit the UHM domain, because trafficking peptides across the cell membrane is challenging. However, the authors do not return to this point in the article.

- Have these compounds be tested in cells? If not, could these compounds be used for studying the splicing process in cells, given that these compounds inhibit most likely all UHM domains?

*Indeed, we have tried to test these compounds in cells. Unfortunately, we did not observe any splicing inhibition in cells up to 100 μ M concentration, while beyond 100 μ M concentration the compounds were toxic to the cells. In fact, the toxicity of phenothiazines at high μ M concentrations has been previously reported (Hawtrey A, et. al. *Low concentrations of chlorpromazine and related phenothiazines stimulate gene transfer in HeLa cells via receptor-mediated endocytosis.* Drug Deliv. 2002;9(1):47-53).*

Therefore, we reverted to in vitro splicing assays, which showed that the splicing inhibition activity is significantly weaker compared to the binding affinity of the inhibitors in vitro. This suggests that the compounds bind to other factors in the nuclear extracts, explaining why we did not observe any inhibitory activity of these compounds on splicing in our cellular assays up to 100 μ M concentration. Therefore, although we know that these compounds enter the cells, currently we are unable to test their effect on splicing in vivo due to toxicity.

- How do these compounds compare to the UHM peptide inhibitors in in vitro splicing reactions?

Like compounds 7-9, our previously reported cyclic peptide inhibitors also inhibit complex A formation. Interestingly, our best cyclic peptide inhibitor (P10) shows a ~300-fold lower inhibitory activity in nuclear extract compared to *in vitro* binding studies to the UHM domains. Although the P10 cyclic peptide is highly selective for the SPF45 UHM domain, it may bind non-specifically to other components of the nuclear extract.

2.) A diagram showing all known UHM-ULM interactions formed during spliceosome assembly and the proposed consequences for spliceosome assembly if these UHM-ULM interactions are inhibited would be useful for the reader.

Thank you for this suggestion, we have now added a diagram in the supplement (**Supplementary Figure S1A**), indicating known UHM proteins in early spliceosome assembly.

3.) The IgM pre-mRNA was used for testing compounds 7 - 9 in *in vitro* splicing reactions. The parent compound however was tested using the MINX pre-mRNA. The type of pre-mRNAs used in *in vitro* splicing reactions can influence the efficiency as well as the effect of a compound on the splicing reactions. To allow a better comparison between the compounds the same pre-mRNA, either IgM or MINX, should be used.

We agree that the splicing of different pre-mRNAs may be differentially affected by the compounds tested here, with some being more sensitive than others. We intentionally tested two different pre-mRNAs with more (MINX) or less (IgM) pronounced (strong) polypyrimidine tracts, with the idea that a substrate whose splicing is potentially more dependent on U2AF, may be more sensitive to the inhibitory compounds. Indeed, the inhibition efficiency with compound 7 was slightly better with the IgM pre-mRNA, where A complex formation was essentially completely abolished at 2 mM; with MINX very low levels of the A complex were still observed at 2 mM. This is now shown in **Supplementary Figure S6**. Therefore, we carried out splicing inhibition assays for compound 7-9 with the IgM pre-mRNA. As compound 8 and 9 have only very minor substitutions relative to compound 7, we expect similar inhibition effects of these compounds on MINX splicing. The parent compound was tested with MINX as for validation of the mechanisms of action we wanted to analyze the cross-exon formation, which is well established with the MINX cross exon construct and has been extensively studied previously.

4.) It is most likely that the additional bands in the splicing reactions containing compound in Figure 6A (running between the MINX pre-mRNA and MINX RNA) are already present in the input RNA. To exclude the possibility that these bands are a result of the treatment, a lane showing the input pre-RNA would be useful.

The best control for differences due to the addition of 7,8-dihydroxyperphenazine is a comparison with the solvent (DMSO), which was also incubated for the same length of time. This control is shown in the first two lanes of **Figure 6A**. These additional bands are not present in the DMSO lanes and their intensity increases with increasing concentrations of 7,8-dihydroxyperphenazine, which supports the conclusion that they result from the presence of this compound.

5.) In the discussion, the authors argue that one reason for 7,8-dihydroxyperphenazine behaving differently to compound 7-9 in splicing reactions, is that "7,8-dihydroxyperphenazine binds to UHM domains involved in splicing complex formation and thereby may trigger the oxidation of labile cysteines in spliceosome associated proteins".

- Would that not be an argument for the spliceosome to be stalled at the E-complex formation?

Thank you for pointing this out. To clarify this point, we suggest that the inhibitory concentration of 50 μ M observed for 7,8 dihydroxyperphenazine is significantly lower compared to the close analogues (cmp 7- cmp 9) in the splicing assays. At this concentration, we only expect a very minor population of UHM domains to be occupied by 7,8 dihydroxyperphenazine, which is not sufficient to inhibit the interaction completely. We therefore suggest that there could be alternative ways how 7,8 dihydroxyperphenazine could inhibit splicing, one being by generating reactive radicals.

We have replaced the original sentence with the following: "It is thus possible that 7,8-dihydroxyperphenazine blocks spliceosome assembly by oxidizing labile cysteines of proteins specifically found in spliceosomal A complexes, or in the U4/U6.U5 tri-snRNP, thereby inhibiting its docking."

- Are there any known UHM-ULM interaction involved in the transition from the A to the B complex?

We are not aware of any specific UHM-ULM interactions involved in the transition from the A to B complex formation. However, there are a number of UHM containing proteins implicated in alternative splicing of specific genes, which could play a role here. As mentioned above, in the presence of 7,8 dihydroxyperphenazine, the A and A-like complexes that still form may be structurally compromised in some way (one possibility being oxidation of cysteines in the A complex), thus precluding B complex formation. Alternatively, 7,8 dihydroxyperphenazine may oxidize cysteines of proteins in the U4/U6.U5 tri-snRNP, preventing it from stably associating with the A or A-like complex.

Minor flaws:

1.) Line 78/79 "Only few inhibitors are known to act during the early phase of spliceosome assembly, i.e. the formation of the early E complex...." - Which inhibitors are known to stall the spliceosome assembly at the E complex formation?

Thank you for this comment. We have now added two references where inhibitors of early spliceosome assembly (H/E complex formation) are described (Effenberger, K. A. et al. 2013 and Pilch, B. et al. 2001).

2.) Line 99: "loss in binding affinity of the probe to the UHM domain"

- SPF45 UHM domain?

We apologize for the confusion. It is the SPF45 UHM domain. We have corrected this in the manuscript.

3.) Line 107: Figure legend 1 "the binding affinity of this interaction is" - The binding affinity of the SPF3B1 ULM - SPF45 UHM interaction?

We meant the interaction between the tracer ([sc,sc(KSRWDE)]-K-C-fluorescein) and the SPF45 UHM domain. We have corrected this in the revised manuscript.

4.) Line 113/114: "at a fixed concentration "

- What concentration?

We have included this information in the manuscript. A fixed compound concentration of 125 μ M was used during screening of the library.

5.) In Figure 6 C the authors perform *in vitro* splicing reactions using "the MINX exon RNA that contains an exon flanked upstream by a branch-site/3'ss and downstream by a 5'ss". The splicing reactions are carried out in the presence and absence of 5'ss oligonucleotides. Are these antisense oligonucleotides to block the 5'ss? These oligos are not mentioned in the material and methods section.

This is in fact not an anti-sense oligonucleotide but a short RNA oligonucleotide that contains a 5' splice site sequence. We now include a more detailed description of the experiments in *Results* and *Materials and Methods*. The mechanism whereby the 5' ss oligo aids the stable interaction of the tri-snRNP is to act as a mimic of a cross-intron upstream 5'ss.

6.) In Figure 6 D and E anacardic acid was used to inhibit tri-snRNPs binding to the A like complex, this however is not mentioned in the text of the results section, only in material and methods.

We apologize for this oversight. We now state in the legend to Figure 6 that "Anacardic acid (AA), which disrupts the U4/U6.U5 tri-snRNP, was used as a positive control to stall assembly at the A-like complex stage".

7.) Supplementary figure 2 legend: SPF45 UHM domain?

We have corrected this in the Supplementary Figure 2 legend.

8.) Supplementary figure 3 legend: SPF45 UHM domain?

We have corrected this in the Supplementary Figure 3 legend

Reviewer #2

This manuscript describes a successful HTS campaign to discover novel inhibitors of the UHM-ULM interaction, which is a central feature of complexes that regulate splicing. The results are compelling, the compounds are important new tools for dissecting splicing, and the work represents a key first step toward generating bioactive inhibitors of splicing as therapeutics.

Minor comments:

1). *How does the Haddock pose compare to the co-crystal structure? This could be included in SI.*

We have added the superposition of the Haddock model of SPF45 UHM domain-inhibitor complex and the crystal structure of PUF60 UHM domain-inhibitor complex in the **Supplementary Figure S5C**.

2). *It is interesting that the more potent compounds show less selectivity among UHM domains. This could be a potential challenge as these domains have conserved structure due to interactions with multiple ULMs. Are there any "exo-sites" that could be used to build in additional selectivity?*

As the reviewer points out, it is indeed not straightforward to achieve UHM selectivity due to the conserved nature of UHM domains. However, there is at least one "exo-site", which can be exploited to design selectivity in these compounds. This is Tyr-376 in the SPF45 UHM domain (**Figure S5C**), which is quite variable in different UHM domains (Trp in PUF60 and Lys in U2AF65). We have shown that this position can be used to discriminate between SPF45 and U2AF65 UHM domains in our previously published study of cyclic peptide inhibitors of UHM-ULM interactions (*Jagtap et.al., J. Med. Chem. 2016, 59, 10190–10197*). Due to difficulties in the chemical synthesis of phenothiazines with substituents, which could target the corresponding position, a similar strategy has so far not been successful for small molecule ligands.

3). *While it is impressive that the compounds are active in splicing extracts, their IC-50 are nearly 3 orders of magnitude larger than the Kd for binding. The authors suggest an abundant reservoir of other binding sites sequestering the compound. There is a danger that the splicing inhibition is due to millimolar inhibitor inducing effects not related to UHM binding. This might be difficult to untangle experimentally, but could be mentioned.*

As the reviewer mentioned, its indeed difficult to untangle the effect of high concentrations of the inhibitor causing some effects not related to UHM binding at this stage. We have now included a cautionary sentence in the manuscript on p. 12 mentioning this possibility.

4). *On this point, it is stated that the inhibitors cannot compete against the U2AF35-U2AF2-65 complex due to the nanomolar affinity. Perhaps, but this would require some knowledge of the concentrations of those components, since two competitors will compete equally if they are present at their respective Kds. Mass action can overcome even a nM interaction.*

Thank you for pointing this out, We agree with the reviewer that this is a relevant point. As far as we know precise cellular concentrations of U2AF are not known, and in principle there could be differences in subcellular concentrations given that these proteins interact with mRNA and additional factors, not even considering concentration effects due to liquid/liquid phase separation. We have decided to remove the sentence.

5). *Presumably, the compounds have been tested in cells...*

In fact, we have tried to test these compounds in cells. Unfortunately, we did not observe any splicing inhibition in cells up to 100 μ M concentration while beyond 100 μ M concentration the compounds were

toxic to the cells. The toxicity of phenothiazines at high μM concentrations has also been previously reported (Hawtrey A, et. al. *Low concentrations of chlorpromazine and related phenothiazines stimulate gene transfer in HeLa cells via receptor-mediated endocytosis. Drug Deliv. 2002;9(1):47-53*).

Overall, this is an impressive study, carefully done with a wide range of techniques applied. These compounds will be powerful tools too build on in probing the mechanism of splicing regulation. Also, it is important to note that protein-protein interactions are notoriously difficult to inhibit with small molecules, and this work is a splendid example of this that can serve as a paradigm.

We thank the reviewer for appreciating our efforts and the study.

Reviewer #3

The manuscript by Jagtap and co-workers discuss the identification of phenothiazines as general inhibitors of the interaction between UHM domains and the cognate ULM peptides, a type of interaction which is key to the assembly of splicing complexes. The authors use a range of techniques, including fluorescence polarization, isothermal titration calorimetry, AlphaScreen assays, NMR, crystallography, molecular modelling and splicing assays to identify phenothiazines as inhibitors of the interaction, characterize the binding and inhibitor activities of different compound of this class and design optimized compounds. They conclude phenothiazines have the potential to become important tools for investigating splicing and alternative splicing, and may be of therapeutic relevance. The manuscript is clearly written and well organized. The Figures are informative and the conclusions are interesting and overall well supported.

We thank the reviewer for overall appreciation of our work.

A few points should be addressed prior to publication:

1) One important question is whether these compounds can target specific UHM-ULM interactions or represent general inhibitors. In vitro, similar EC_{50} is observed for the different compounds. However, the authors observe a specificity of action in the splicing assays performed. They discuss two possible causes. Clearly, it is very difficult to give a full account of the actions on the compounds on splicing at this stage, but the paper would very much benefit if the two possibilities were explored.

We agree that delineating the exact mechanism of inhibition of 7,8 dihydroxyperphenazine during splicing assays in nuclear extracts will be very useful. Towards this goal we attempted synthesis of biotinylated versions of our compounds to pull-out, which proteins (besides UHM domains) 7,8 dihydroxyperphenazine binds to in the splicing extracts. However, the chemical synthesis was not successful. Besides this, we also explored the inhibitory effects of completely oxidized 7,8 dihydroxyperphenazine, but did not observe any difference compared to “non-oxidized” 7,8 dihydroxyperphenazine. As the reviewer appreciates, this is indeed a difficult challenge and we would continue to explore this in further detail in the future. However, we believe that this is beyond the scope of the current manuscript.

2) The authors mention “...some inhibitors during the early phase of spliceosome...” in lines 78-80. This work must be referenced, this information is important to place the results in the context of the literature.

We have now added two references where inhibitors of early spliceosome assembly (H/E complex formation) are described, namely Effenberger, K. A. et al. (2013) and Pilch, B. et al. (2001).

3) Does the cyclic peptide discussed in lines 96 to 101 bind like an ULM peptide? The authors refer to previous work - could they elaborate on this statement detailing the evidence presented in that paper? Is additional experimental evidence, e.g. NMR, required here? It is unclear at the moment.

Yes, the cyclic peptide discussed in lines 96 to 101 binds like the native ULM peptide as it has been derived from the native ULM peptide of SPF45 UHM domain. The mode of binding (using NMR) and affinity of this peptide has been confirmed in our previous publication (Jagtap et al *J. Med Chem*, 2019). We have added a few sentences clarify this point in the main text.

4) In the Methods the authors mention they titrate the two compounds into the protein and record NMR experiments. Did they only use a 1:1 ration? It would be interesting to know ratios they used for the titration and show the spectra in the Supplementary Materials? Also, it would be interesting to see a binding curve.

Yes, we have used a 1:1.6 ratio of the protein to inhibitor for the single point NMR titration shown in **Figure 3** to show the chemical shift changes of the signals at comparable conditions, suggesting that the two inhibitors bind with similar affinity. We have now performed NMR titration experiments, following NMR signals as a function of ligand concentrations. **Figure 3A** shows a comparison of the ligand bound spectra at 1.6 molar excess of the two ligands compared to the protein. We included the NMR titration steps and the binding curve derived from NMR titration in the supplementary information along with NMR-derived K_D for these compounds (**Supplementary Figure S2**). The methods section has been updated to describe the experiments and data analysis for these new data.

5) Again, in the Methods, the authors mention a 2:1 ratio is used to record two of the NMR experiments. Has this ratio be used for other experiments? Maybe the authors can spell out the rationale of this choice – some of the readers less experienced in this type of studies may wonder.

We apologize for this confusion. The purpose of using a 2-fold excess of compound compared to protein in the 2D and 3D ω_1 -filtered NOESY experiments was to completely saturate the protein so as to avoid signal from the free protein. The rationale for this has now been added in the methods section.

6) Line 185. *Crystal structure of PUF60 UHM domain-inhibitor complex in bold.*

We have corrected this in the manuscript.

7) Line 219. *Structure-based optimization of 7,8-dimethoxyperphenazine in bold.*

We have corrected this in the manuscript.

8) *Replace supplementary table s1 by Supplementary Table S1 (line 373).*

We have corrected this in the manuscript.

9) *Change the font in lines 449, 450, 451, 455, 463, and 471.*

The font issues (small and not bold at some places) apparently appears during the conversion of the manuscript from Word to PDF in the submission system, and will try to avoid this in the resubmission.

REVIEWERS' COMMENTS

Reviewer #1 (Remarks to the Author):

I am satisfied that the authors have addressed all my comments, and I consider the paper ready for publication in Nature Communications.

Reviewer #3 (Remarks to the Author):

The authors have addressed my queries